# 0.1% RGN-259 (Thymosin ß4) Ophthalmic Solution Promotes Healing and Improves Comfort in Neurotrophic Keratopathy Patients in a Randomized, Placebo-Controlled, Double-Masked Phase III Clinical Trial

**DOI:** 10.3390/ijms24010554

**Published:** 2022-12-29

**Authors:** Gabriel Sosne, Hynda K. Kleinman, Clark Springs, Robert Hollis Gross, Jihye Sung, Shinwook Kang

**Affiliations:** 1Kresge Eye Institute, Detroit, MI 48201, USA; 2ReGenTree, LLC, Princeton, NJ 08540, USA; 3Eye Specialists of Indiana, Indianapolis, IN 46260, USA; 4Cornea and Cataract Consultants of Arizona, Phoenix, AZ 85032, USA; 5HLB Therapeutics Co., Ltd., Seoul 13554, Republic of Korea

**Keywords:** neurotrophic keratopathy, thymosin ß4, clinical trial, corneal healing, ocular discomfort

## Abstract

We determined the efficacy and safety of 0.1% RGN-259 ophthalmic solution (containing the regenerative protein thymosin ß4) in promoting the healing of persistent epithelial defects in patients with Stages 2 and 3 neurotrophic keratopathy. Complete healing occurred after 4 weeks in 6 of the 10 RGN-259-treated subjects and in 1 of the 8 placebo-treated subjects (*p* = 0.0656), indicating a strong efficacy trend. Additional efficacy was seen in the significant healing (*p* = 0.0359) with no recurrent defects observed at day 43, two weeks after cessation of treatment, while the one healed placebo-treated subject at day 28 suffered a recurrence at day 43. The Mackie classification disease stage improved in the RGN-259-treated group at Days 29, 36, and 43 (*p* = 0.0818, 0.0625, and 0.0467, respectively). Time to complete healing also showed a trend towards efficacy (*p* = 0.0829, Kaplan–Meier) with 0.1% RGN-259. RGN-259-treated subjects had significant improvements at multiple time points in ocular discomfort, foreign body sensation, and dryness which were not seen in the placebo group. No significant adverse effects were observed. In summary, the use of 0.1% RGN-259 promotes rapid healing of epithelial defects in neurotrophic keratopathy, improves ocular comfort, and is safe for treating this challenging population of patients.

## 1. Introduction

Neurotrophic keratopathy (NK) is a rare and devastating eye disease associated with damage to either the trigeminal nerve or its branches resulting in epithelial defects on the surface of the cornea [1,2]. Patients with the severe/progressed forms of NK can have permanent vision loss. The most common causes are the herpes simplex 1 virus, the herpes zoster virus, and surgery involving the cornea [3]. NK has a prevalence estimated as 21.34 cases per 100,000 patients in the USA [4]. In the early stage of the disease (Stage 1), there is increased viscosity of the tear mucus, superficial punctate keratopathy, and epithelial hyperplasia, which can be treated symptomatically with both artificial tears and antibiotics [3,5]. In the second stage (Stage 2), the damage to the ocular surface is more severe with stromal swelling, epithelial defects, and loosening of the epithelium. Stage 2 disease can be treated with either bandage contact lenses, amniotic membrane, conjunctival flaps, tarsorrhaphy, or recombinant nerve growth factor (NGF, Oxervate™). In the final stage (Stage 3), a corneal ulcer, stromal lysis, corneal scarring, and perforation can occur with these complications resulting in loss of visual function. Treatment at this stage involves therapeutic contact lenses, amniotic membrane, conjunctival flaps, tarsorrhaphy, cyanoacrylate glue for perforations, keratoplasty, and/or recombinant NGF (Oxervate™). Recombinant NGF (Oxervate™), the only FDA-approved treatment for NK, has shown efficacy in healing the epithelial defect in 65–72% of Oxervate^TM-^-treated patients compared to vehicle-treated patients (16.7–33.3%) when used topically six times a day for 8 weeks [6,7].

Thymosin beta 4 (Tß4), a naturally occurring 43 amino acid-containing protein, is present in both cells and body fluids including tears and has shown efficacy in treating surface ocular wounds in both animal models and patients [8,9]. It has multiple biological activities that promote the healing process. Tß4 promotes cell migration, cell survival, stem cell recruitment, production of laminin-332 (important in migration and cell–cell and cell-matrix contacts), and cytoprotection by reducing oxidative stress, inflammation, and cell death (apoptosis) [8]. Using a synthetic version of the natural molecule, many studies using alkali- or heptanol-treated eyes in both mice and rats show a fast-healing effect with decreased inflammatory cell infiltration [9,10,11]. Furthermore, in an animal model of dry eye, Tß4 was shown to increase surface epithelial healing, improve corneal smoothness, prevent epithelial detachment, increase goblet cell numbers and mucin production, and reduce inflammation [12]. With regard to the underlying pathology in NK, it is important to note that Tß4 has efficacy in the repair and regeneration of nervous system damage in animal models. For example, systemic Tß4 repairs and restores functional activity in damaged brains, including animal models of multiple sclerosis, traumatic brain injury, stroke, and spinal cord injury [13,14,15,16]. When taken together, these animal studies provided the scientific rationale for using Tß4 to treat NK where there is nerve damage resulting in surface corneal epithelial lesions. In a six-patient physician-sponsored, compassionate use study of NK patients, complete healing of the defects was seen in four of the patients at 28 days and in the other two patients by days 55 and 60 with 0.1% RGN-259 (0.1% Tß4 topical ophthalmic solution) which contains the synthetic version of Tß4 as the active drug substance [17]. In addition, significant improvements in both surface healing and discomfort were reported [17]. Subsequent phase II and III clinical studies of moderate to severe dry eye also supported the efficacy of Tß4 in healing the ocular surface damage and in improving dye eye symptoms ([18], unpublished observations). Such activities observed in animal models and in clinical trials demonstrate that Tß4 has the potential to provide clinically meaningful improvements for patients with both dry eye and NK as well as for patients with other ocular surface lesions and pathologies [19].

Here, we describe a phase III study, SEER-1, of 0.1% RGN-259 (timbetasin acetate) topical ophthalmic solution applied five times per day in patients with Stages 2 and 3 neurotrophic keratopathy. The placebo group received the vehicle of RGN-259 used previously in other human ocular studies [17]. The findings included fast and complete healing by 4 weeks in the treated group vs. placebo group. There were no safety issues. Many secondary endpoints, including improved ocular comfort, also demonstrated the beneficial effects of 0.1% RGN-259 for NK.

## 2. Results

### 2.1. RGN-259-Treated Subjects Showed Faster Healing at 4 Weeks

Eighteen subjects were enrolled and completed the study with minimal protocol deviations. At Baseline, the subjects in the placebo group tended to be older (average placebo vs. treatment of 72.5 vs. 63.7 years) and have a larger area of defect (median placebo vs. treatment of 7.375 vs. 6.570 mm^2^) than that in the treated group but both groups were similar in their Mackie Score distribution (Table 1).

The primary efficacy measure was percent of subjects achieving complete healing at Visit 5 (Day 29) of treatment (Table 2). At Visits 5 and 6 (Days 29 and 36), 6 out of 10 subjects (60%) in the 0.1% RGN-259 treatment group achieved complete healing of their epithelial defects compared to 1 out of 8 subjects (12.5%) in the placebo group resulting in a notable difference in the percentage of study eyes with complete healing after treatment with 0.1% RGN-259 compared to placebo near 50% (47.5%). Due to the low sample size, high power was unattainable and achieving statistical significance was difficult. The primary efficacy endpoint, the percentage of subjects achieving complete healing of the persistent epithelial defect as determined by corneal fluorescein staining at 4 weeks (Visit 5) in the ITT population, was not met with statistical significance (*p* = 0.0656) but showed a strong efficacy trend. The *p*-value was obtained from a Fisher’s exact test, which is known to be a conservative test appropriate for small sample sizes. The *p*-value obtained from an ad hoc Chi-square test for larger sample sizes is 0.0400; however, the assumption of larger sample size is violated. The corresponding two-sided asymptotic normal 95% confidence interval (CI) is (9.5%, 85.5%) which excludes 0 and was computed as part of the planned analyses. This provides further evidence of a meaningful difference in the complete wound healing between 0.1% RGN-259 and placebo. In support of this strong efficacy trend, 2 weeks after cessation of treatment (day 43), there was a significant difference in complete healing (95% CI, 19.0%-81.0%; *p* = 0.0359) in favor of 0.1% RGN-259. Additionally, time to complete healing tended to be faster for study eyes treated with RGN-259 than those eyes treated with placebo with near significance (*p* = 0.0829, Kaplan–Meier analysis). Complete healing first occurred at a minimum of 15 days in the 0.1% RGN-259 treatment group while complete healing did not occur in the placebo group until a minimum of 22 days. Additionally, an ad hoc analysis of the epithelial defect area based on an odds ratio adjusted for baseline area of the defect with a logistic regression, showed that subjects in the 0.1% RGN-259 group were roughly 18 times (*p* = 0.0737) more likely to have complete healing in the epithelial defect at Visit 5 (Day 29) than the subjects in the placebo group. Similar to other analyses, the *p*-value was not significant likely due to the small sample size. The odds ratio, roughly 18, is very high with a wide confidence interval. In summary, not only is healing more complete but it is also faster in the 0.1% RGN-259 treated group with significant healing maintained 2 weeks after cessation of treatment.

### 2.2. RGN-259-Treated Subjects Had a Lower Mackie Classification by 4 Weeks

Clinically important shifts to either a lower Mackie classification (Table 3 and Appendix A) or complete healing were consistently demonstrated in the 0.1% RGN-259 treatment group over the course of the study. Specifically, at Visit 5 (Day 29), 80% of the subject eyes treated with 0.1% RGN-259 either shifted to stage 1 Mackie classification of epithelial defects or were completely healed compared to baseline versus only 25% demonstrating similar shifts or healing in the placebo group. A similar result was also seen at one and two weeks after treatment ended (Visit 6 and Visit 7) with 7 subjects (70%) treated with 0.1% RGN-259 shifting to either a lower Mackie classification of stage 1 or complete healing and only 2 subjects (25%) in the placebo group shifted to a lower Mackie classification.

### 2.3. RGN-259-Treated Subjects Had Improved Ocular Discomfort by 2 Weeks

Additional secondary endpoints including ocular discomfort and symptoms were assessed at all scheduled visits using the subjective Ora Calibra™ Ocular Discomfort and 4-Symptom Questionnaire. Ocular discomfort and symptoms were summarized at each visit using quantitative summary statistics. Change from baseline (Day 1) was also summarized for all post-baseline visits. Statistical comparisons were performed using the two-sample *t*-test as the primary method of analysis for 0.1% RGN-259 compared to placebo. A Wilcoxon rank sum test for data at each visit and an ANCOVA model adjusting for baseline for changes from baseline was also assessed.

Differences in improvement from baseline between groups in favor of 0.1% RGN-259 were demonstrated in ocular discomfort and in two of the four symptoms, foreign body sensation and dryness, at multiple visits (Table 4). Some of the observed differences are due to baseline differences between the 0.1% RGN-259 and placebo groups; therefore, both a two-Sample *t*-test and ANCOVA *p*-values are reported. Notable improvements from baseline in favor of treatment with 0.1% RGN-259 where both two-Sample *t*-test and ANCOVA *p*-values were less than 0.05 were seen with the exception of photophobia and burning.

### 2.4. RGN-259 Was Very Safe for the NK Subjects

Sixteen Adverse Events (AEs) were experienced across seven subjects reporting at least one AE (Table 5 and Table 6). Only one of the AEs, experienced by one subject in the 0.1% RGN-259-treated group, was determined to be treatment related. Most of the AEs were ocular (11 events in 18 subjects). Only one non-ocular unrelated Serious Adverse Event (SAE) was reported in the 18 study subjects. A majority of the AEs were classified as mild in severity with only one AE (non-ocular) classified as moderate. None of the AEs occurring over the course of the study resulted in subject withdrawal. By the end of the study, all of the subjects experiencing AEs had recovered and the AEs had been resolved.

Other parameters of safety showed little or no change with 0.1% RGN-259 treatment. In terms of corneal sensitivity, no differences were seen between the 0.1% RGN-259 and placebo treatment groups. There were no other abnormalities or complications on anterior or in the dilated fundoscopy exam in either the 0.1% RGN-259 or placebo treatment groups for Visits 5 and 7. Slit-lamp biomicroscopy revealed some shifts from baseline in both the RGN-259-treated and placebo-treated subjects but these shifts were not significant. There were negligible differences in the group mean visual acuity (logMAR) change from baseline scores in the 0.1% RGN-259 and placebo treatment. Intraocular pressure levels were comparable throughout the study and not clinically significant among treatments. These findings confirm the safety of topical 0.1% RGN-259 for NK patients.

## 3. Discussion

Overall, a strong healing efficacy of 0.1% RGN-259 was demonstrated in this study as evidenced by improvements from baseline in the treatment group differences for both primary and secondary endpoints with excellent safety parameters, including complete healing, rate of healing, lesion size reduction, Mackie classification, and improved comfort measurements. While the primary efficacy endpoint was narrowly missed in this study with statistical significance (*p* = 0.0656), alternative ad hoc Chi-square analysis of the primary efficacy endpoint demonstrated notable differences favoring treatment with 0.1% RGN-259 over placebo (*p* = 0.0400). Given the difficulty enrolling in this rare patient population, the data from this study suggest a clinically meaningful difference in wound healing between 0.1% RGN-259 and placebo that may be confirmed with statistical significance in a larger sample-sized study.

A dose of 0.1% RGN-259 appears to be a safe and well tolerated treatment. RGN-259 contains a synthetic copy of the naturally occurring molecule Tß4 that is normally present in tears [20]. The level in tears decreases with age. A previous trial in severe to moderate dry eye demonstrated the safety of 0.1% RGN-259 [18]. The safety of 0.1% RGN-259 was confirmed in this NK study where a total of 16 AEs was reported in all subjects who had received at least one dose of the study drug. Most of the AEs were ocular and mild. Only one non-ocular SAE was reported and was determined to be unrelated to the study drug. None of the AEs occurring over the course of the study resulted in deaths or in subject withdrawal, and a majority of the AEs were resolved or resolving by the end of the study. One ocular AE involving a corneal epithelium defect was classified as moderate in severity but was classified as unrelated to the study-drug and was resolved by the end of the study. Five ocular AEs were reported in the study eyes of two subjects in the 0.1% RGN-259 treatment group. The other six ocular AEs were reported in the untreated fellow eyes. No significant differences were noted in the dilated fundoscopy, BCVA, and corneal sensitivity with some minimal shifts observed in the slit-lamp biomicroscopy from baseline. These differences were not notable between treatment with 0.1% RGN-259 and placebo. In conclusion, 0.1% RGN-259 was demonstrated to be a safe ophthalmic product for use in the treatment of NK subjects.

Subject reported satisfaction is an important clinically meaningful outcome and can influence compliance with the medication. In this study, not only does 0.1% RGN-259 promote faster healing but it also increases ocular comfort and reduces foreign body sensation and dryness which is a strong benefit for the patient. Furthermore, in a previous small open label study of 4 NK patients treated with 0.1% RGN-259, no subjects reported discomfort with the eye drops but did report improved ocular comfort [17]. It was unexpected that comfort measures would be improved in NK patients treated with 0.1% RGN-259 since reduced sensitivity is a hallmark of NK [3]. Part of the explanation for the improved comfort could be due to the fast action of 0.1% RGN-259 in healing the defect and normalizing the ocular surface. This is likely due to its multiple mechanisms of action beyond promoting corneal epithelial cell migration to repair the epithelial defect [8]. In animal models, 0.1% RGN-259 has been shown to reduce inflammation which can cause pain, burning, itching, etc. as well as contribute to the cellular/stromal damage [3,18]. In an animal model of dry eye, Tß4 promoted corneal integrity in part through increases in laminin-332 production [9]. Such activity counteracts many of the issues with lifting of the epithelium in NK. Finally, in an animal model of dry eye, it has been shown to increase goblet cell numbers and mucin production which are important to creating a normal tear film [3,12]. Such activities in “normalizing” the corneal epithelium are expected to contribute in part to the improved comfort of the NK patients receiving 0.1% RGN-259.

It is not known if RGN-259 helps to repair the nerve damage in NK. In animal models, when given systemically, it has clearly been shown to recruit stem cells to repair brain injury and to regenerate nerves due to damage from trauma, multiple sclerosis, and stroke [13,14,15,16]. Tß4 has also been shown by proteomic analysis to increase endogenously in an optic nerve crush rat model 3 days after injury when the cells begin to die [21]. When given systemically, Tß4 not only increased retinal ganglion survival 3-fold over untreated animals but also promoted axon regeneration. These data suggest that in NK, RGN-259 might also be acting by promoting neuronal cell survival and/or nerve regeneration. Studies in an NK animal model will be needed to determine whether RGN-259 can repair nerve damage in NK. This study showed no difference in corneal sensation between the two groups which may indicate no effect on nerve growth with topical use. Further studies are necessary with confocal microscopy to demonstrate the effect of this agent on nerve regrowth.

Scar formation is a severe and debilitating consequence of NK that leads to blindness [1,2,3]. Systemically given Tß4 also reduces scar formation/fibrosis in many tissues (skin, heart, lungs, kidney, and liver) with scarring due to varying underlying causes [22,23,24,25,26,27]. It has not yet been tested for reduction in scar formation in the eye. In a bile duct ligation animal model of liver fibrosis, Tß4 reduced the expression of many molecules important in the fibrotic pathway including TGF-ß1 (Transforming growth factor-ß1), TGF-ß RII, Smad2, and Smad3 [22]. It acts to reduce scar formation by decreasing the number of myofibroblasts and allowing better alignment of collagen fibers. Reducing the inflammation also helps to prevent scar formation. The first 4 amino acids of Tß4, SDKP (serine-aspartate-lysine-proline), occur naturally in tissues and body fluids and have anti-inflammation and antifibrotic activity [28,29,30,31,32,33,34]. In animal models of fibrotic diseases, Ac-SDKP has anti-fibrotic activity in the heart, liver, and kidney. For example, Ac-SDKP not only prevented cardiac fibrosis in rat models with renovascular hypertension but also reversed the fibrosis [29]. Ac-SDKP acts by reducing inflammation and TGF-ß1 levels and likely will be beneficial to the NK patients in preventing and reducing scarring and possibly in reversing the scarring. It will be important in future studies to assess the effects of 0.1%RGN-259 on reducing scar formation in NK patients.

The limitations of this study were the small number of subjects with 10 in the treated group and 8 in the placebo group. The increased average age and size of the lesion in the placebo population may also have contributed to the reduced rate of healing. Because NK is a rare disease, many sites were needed to recruit the subjects further adding to the problem of data uniformity. In addition, the ocular discomfort analyses were based on subjective information. This information was provided by the subject at each office visit in consult with professional staff and thus helped to provide more reliable and consistent information. Despite these limitations, the data showed a strong trend for healing and significant findings for the subjects in many important parameters, indicating that RGN-259 was fast acting and has the potential to yield clinically meaningful findings in a larger patient population.

Oxervate™ (recombinant nerve growth factor, NGF) is the only approved treatment for NK, and it works well when given six times a day for 8 weeks [6]. It is, however, costly and not reimbursed completely for all patients and the length of treatment makes compliance difficult. Oxervate™ requires cold chain delivery and storage. Less costly treatments, such as tarsorrhaphy, conjunctival flap, sutured and sutureless amniotic membrane transplantation, and blood derived products, are available but some have limited efficacy. 0.1% RGN-259 may be an alternative treatment for NK due to its faster healing properties, lower cost, use at room temperature, and potential increased patient comfort.

## 4. Materials and Methods

A multicenter prospective, randomized study, registered at clinicaltrials.gov as NCT02600429, was performed in the USA in accordance with the tenets of the Declaration of Helsinki. The participating sites included The Hull Eye Center, Lancaster, CA; Vision Institute, Colorado Springs, CO; Eye Center of Northern Colorado, Fort Collins, CO; Insight Vision Group, Parker, CO; Medical Faculty Associates Inc, Washington, DC; Midwest Cornea Associates, LLC, Indianapolis, IN; Koffler Vision Group, Lexington, KY; Richard Eiferman, MD, PSC, Louisville, KY; Central Maine Eye Care, Lewiston; ME, Black Hills Regional Eye Institute, Rapid City, SD; Michigan Cornea Consultants, Southfield MI; Glaucoma Consultants of Colorado, Parker CO; and Cornea and Cataract Consultants of Arizona, Phoenix, AZ. The placebo and 0.1% RGN-259 were prepared as previously described. The study was conducted by Ora Inc, Andover, MA. The protocol and its amendments, informed consent form and assent form, Health Insurance Portability and Accountability Act (HIPAA) form, print advertisement, screening and enrollment form, primary care physician notification form, and subject diary instructions were reviewed by a properly constituted IEC or IRB (Alpha, IRB, San Clemente, CA, USA).

Subject demographics: Ten subjects were in the treatment group and eight were in the placebo group. Subjects were enrolled from multiple centers and their demographics and clinical characteristics are shown in Table 1. Originally, a study with 46 subjects treated in a 2:1 ratio of 0.1% RGN-259 vs. Placebo was planned but because of slow recruitment for this rare disease, the study was terminated early after 18 subjects had completed the study. Informed consent was obtained from all subjects involved in the study.

Study design: The primary outcome measure was percentage of subjects achieving complete healing of the persistent epithelial defect (PED) as determined by corneal fluorescein staining at Day 29 after the first dosing. The secondary outcome measures (all after first dosing) were (1) Percentage of subjects achieving complete healing at Days 8, 15, 22, 36, and 43, (2) Epithelial Defect Measurement (3) Classification as stage 1, 2 or 3 using Mackie Classification at Days 8, 15, 22, 29, 36, and 43, (4) Tear Film Break-up Time at Days 29, 36, and 43, (5) Ocular Discomfort (Ora Calibra™ Ocular Discomfort and 4-Symptom Questionnaire, a 6-point scale (0–5) on which 0 is no pain and 5 is the worst and in which five ocular symptoms are individually graded: overall discomfort, photophobia, foreign body sensation, burning, and dryness) at Days 8, 15, 22, 29, 36, and 43, and (6) Visual acuity (LogMAR using an Early Treatment Diabetic Retinopathy Study scale (ETDRS) chart) at Days 8, 15, 22, 29, 36, and 43.

Safety outcome measures all after the first dosing were (1) Visual acuity (LogMAR) at Days 8, 15, 22, 29, 36, and 43, (2) Change in anterior segment biomicroscopy using slit-lamp at Days 8, 15, 22, 29, 36, and 43, (3) Corneal Sensitivity using the aesthesiometer (Cochet-Bonnet) at Days 8, 15, and 29, (4) Adverse event query at Visits at Days 8, 15, 22, 29, 36, and 43, (5) Change in posterior segment biomicroscopy using Dilated Fundoscopy at Days 29 and 43, and (6) Intraocular Pressure at Days 29 and 43.

The study involved 7 Visits with assessments as shown in Figure 1. A total of 18 subjects met the Inclusion Criteria which were: male or female of any race, at least 18 years of age, had provided verbal and written informed consent, had a persistent epithelial defect of at least 2 mm in length and confirmed not to be simply superficial punctate keratitis at Visit 1 for at least 1 week with failure of conventional, nonsurgical treatment, had evidence of decreased corneal sensitivity ≤ 4.5 cm using the Cochet-Bonnet aesthesiometer at Visit 1, were able and willing to follow instructions, including participation in all study assessments and visits, had Stage 2 or 3 neurotrophic keratopathy in at least one eye, and if a female of childbearing potential, had a negative urine pregnancy test at Visit 1 (Day 1) and agreed to use an adequate method of birth control throughout the study period. The Exclusion Criteria included: have any clinically significant slit lamp findings at Visit 1 (Day 1) that in the opinion of the investigator may interfere with the study parameters, have significant blepharitis, meibomian gland dysfunction, lid margin inflammation or active ocular allergy that requires treatment, have a lid function abnormality which, in the opinion of the investigator, is the primary cause of the persistent epithelial defect, be diagnosed with ongoing ocular infection (bacterial, viral or fungal) or active inflammation not related to NK, anticipate the use of fluoroquinolone-containing antibiotic eye drops during the study, have used contact lenses (excluding therapeutic contact lenses) within 14 days prior to Visit 1 (Day 1) or anticipates use of contact lenses during the study period, have an uncontrolled systemic disease that in the opinion of the investigator may interfere with the study parameters, and anticipate a change in immunosuppressive therapy during the course of the study. For patients with bilateral neurotrophic keratopathy, the study eye from each subject was selected based on the affected eye with the largest affected area at baseline. Safety measurements included Visual Acuity (LogMAR), Slit-Lamp Biomicroscopy, Corneal sensitivity (Cochet-Bonnet), Adverse Event Query, Dilated Fundoscopy, and Intraocular Pressure.

Statistical Analyses: All data analyses were performed by the SDC after the study was completed and the database had been locked and released for unmasking. Statistical programming and analyses were performed using SAS^®^ Version 9.4 (Cary, CA, USA). All statistical tests are two-sided with a significance level of 0.05 (α = 0.05). All CIs are two-sided at 95% confidence.

The number of study eyes achieving complete healing of the epithelial defect at each scheduled visit was summarized using frequencies and percentages, as well as a 95% asymptotic normal CI for each treatment group. The primary efficacy endpoint of the percentage of study eyes achieving complete healing of the epithelial defect as measured by corneal fluorescein staining at Week 4 (Visit 5) was performed on the Intention-To-Treat (ITT) population, with subjects who withdraw early, used escape medications, or had escaped surgical interventions imputed as not having resolution of the epithelial defect. The primary endpoint was tested using a two-sided Fisher’s exact test with an alpha of 0.05. Additionally, a 95% asymptotic normal CI around the difference in percentages (RGN-259—placebo) was calculated.

The continuous and ordinal secondary efficacy variables collected at each visit were analyzed for the ITT population only either with a two-sample *t*-test or Fisher’s Exact Test comparing the active treatment group to placebo. All visit-based data were analyzed at each visit as well as for changes from baseline (Day 1, Visit 1). A Wilcoxon rank sum test was also assessed where appropriate. Sensitivity analyses were performed on the percentage of study eyes with complete healing of the primary defect at Week 4 (Visit 5). The sensitivity analysis was performed for both the ITT and Per-protocol (PP) populations on observed data only, where only subjects having data at Week 4 (Visit 5) were included.

A logistic regression model was used to test if the proportion of study eyes of the ITT population with complete healing differed between treatment groups when adjusted for the subjects’ baseline defined at Visit 1 approximate area of the defect. The approximate area was calculated as the defect length × defect width. All follow-up visits (Visits 2, 3, 4, 5, 6 and 7) were tested with Visit 5 (Day 29) being the primary visit of analysis.

Corneal sensitivity was analyzed in the same manner as epithelial defects. Specifically, the continuous and ordinal variables collected at each visit were summarized descriptively (n, mean, standard deviation (SD), median, min and max), and were analyzed with two-sample *t*-test, Wilcoxon rank sum test, and Analysis of Covariance (ANCOVA), where appropriate, comparing the active treatment group to placebo. All visit-based data were analyzed at each visit as well as for changes from baseline on the Safety population.

A Chi-square test of statistical significance utilizing an asymptotically normal distribution was used as an alternative analysis on the primary efficacy endpoint using the same primary analysis variables and the ITT population.

## Figures and Tables

**Figure 1 ijms-24-00554-f001:**
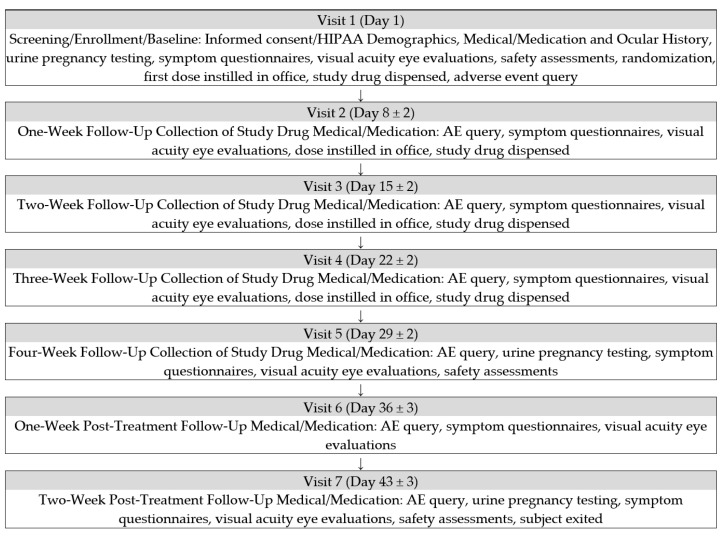
Study Flow Chart.

**Table 1 ijms-24-00554-t001:** Subject Demographics and Clinical Characteristics.

Variable	Treated with 0.1% RGN-259 (*n* = 10)	Placebo (*n* = 8)
Gender (*n*)		
Male	2	4
Female	8	4
Average Age (SD)	63.7 (15.58)	72.5 (7.87)
Race		
Hispanic Latino	0	1
White	10	7
Mackie Classification		
Stage 1	0	0
Stage 2	9	7
Stage 3	1	1
Frequent Eye Disorders		
Cataract	7	8
Corneal opacity	0	3
Corneal scar	1	2
Dry eye	9	6
Glaucoma	2	2
Open angle glaucoma	2	1
Retinal detachment	1	3
Surgical Procedures		
Cataract	7	4
Eye surgery	1	2
Lens implant	4	4
Keratoplasty	2	3
Post lens Capsulotomy	2	2
Retinal laser coagulation.	2	2
Retinopexy	1	2
Area of Epithelial Defect (mm^2^)		
Mean (SD)	6.815 (3.848)	9.871 (7.134)
Median	6.570	7.375
Duration of NK (Days)		
Mean (SD)	213 (229.2)	973 (1557.3)
Median	107.5	206.0
Ocular Discomfort *		
Mean (SD)	3.3 (1.16)	1.6 (1.60)
Median	3.0	1.5
Visual acuity (log MAR)		
Mean (SD)	1.386 (1.1566)	1.196 (0.9098)

* Assessed by Ora Calibra™ Ocular Discomfort and 4-Symptom Questionnaire.

**Table 2 ijms-24-00554-t002:** Primary Endpoint: Healing of Epithelial Defect in the ITT Population.

Visit Measurement	0.1% RGN-259 (*n* = 10)	Placebo (*n* = 8)
**Visit 2 (Day 8 ± 2)**		
Complete Healing: *n* (%)	0	0
Two-sided 95% CI	(0.000, 0.000)	(0.000, 0.000)
Difference in Proportions, RGN-259—Placebo	0.000	-
Two-sided 95% CI	(NC, NC)	-
*p*-value, Fisher’s Exact Test (Primary)	NC	-
*p*-value, Chi-square Test (Ad Hoc)	NC	
**Visit 3 (Day 15 ± 2)**		
Complete Healing: *n* (%)	3 (30.0%)	1 (12.5%)
Two-sided 95% CI	(0.016, 0.584)	(0.000, 0.354)
Difference in Proportions, RGN-259—Placebo	0.175	-
Two-sided 95% CI	(−0.190, 0.540)	-
*p*-value, Fisher’s Exact Test (Primary)	0.5882	-
*p*-value, Chi-square Test (Ad Hoc)	0.3749	
**Visit 4 (Day 22 ± 2)**		
Complete Healing: *n* (%)	4 (40.0%)	2 (25.0%)
Two-sided 95% CI	(0.096, 0.704)	(0.000, 0.550)
Difference in Proportions, RGN-259—Placebo	0.150	-
Two-sided 95% CI	(−0.277, 0.577)	-
*p*-value, Fisher’s Exact Test (Primary)	0.6380	-
*p*-value, Chi-square Test (Ad Hoc)	0.5023	
**Visit 5 (Day 29 ± 2)**		
Complete Healing: *n* (%)	6 (60.0%)	1 (12.5%)
Two-sided 95% CI	(0.296, 0.904)	(0.000, 0.354)
Difference in Proportions, RGN-259—Placebo	0.475	-
Two-sided 95% CI	(0.095, 0.855)	-
*p*-value, Fisher’s Exact Test (Primary)	**0.0656**	-
*p*-value, Chi-square Test (Ad Hoc)	**0.0400**	
**Visit 6 (Day 36 ± 3)**		
Complete Healing: *n* (%)	4 (40.0%)	1 (12.5%)
Two-sided 95% CI	(0.096, 0.704)	(0.000, 0.354)
Difference in Proportions, RGN-259—Placebo	0.275	-
Two-sided 95% CI	(−0.105, 0.655)	-
*p*-value, Fisher’s Exact Test (Primary)	0.3137	-
*p*-value, Chi-square Test (Ad Hoc)	0.1955	
**Visit 7 (Day 43 ± 3)**		
Complete Healing: *n* (%)	**5 (50.0%)**	**0**
Two-sided 95% CI	(0.190, 0.810)	(0.000, 0.000)
Difference in Proportions, RGN-259—Placebo	0.500	-
Two-sided 95% CI	(0.190, 0.810)	-
*p*-value, Fisher’s Exact Test (Primary)	**0.0359**	-
*p*-value, Chi-square Test (Ad Hoc)	**0.0186**	

Abbreviations: CI = Confidence Interval; ITT = intent-to-treat; NC = Not Calculable. The bold is to emphasize the statistically significant data.

**Table 3 ijms-24-00554-t003:** Change in Mackie Score with Treatment over Time.

	Day 1		Day 29		Day 36		Day 43	
	0.1% RGN-259	Placebo	0.1% RGN-259	Placebo	0.1% RGN-259 *	Placebo	0.1% RGN-259 *	Placebo
Stage 1	0	0	6	2	5	1	4	1
Stage 2	9	7	2	6	3	6	3	7
Stage 3	1	1	0	0	0	1	0	0
*p* value			0.0818		0.0625		**0.0467**	

* Two subjects in the treated group had no detectable NK at Days 29 and 36, and three subjects in the RGN-259-treated group at Day 43 had no detectable NK by the Mackie classification.

**Table 4 ijms-24-00554-t004:** Ora Calibra™ Ocular Discomfort and 4-Symptom Questionnaire Change from Baseline, Visits 2, 3, 4, 5, 6, and 7.

Visit	Statistical Measurement	0.1% RGN-259 (*n* = 10)	Placebo (*n* = 8)
**Ocular Discomfort**			
Visit 2 **(Day 8 ± 2)**	Mean (SD)	−1.5 (0.97)	−0.6 (1.19)
	*p*-value, Two-Sample *t*-test	0.1045	-
	*p*-value, ANCOVA model	0.6676	-
Visit 3 **(Day 15 ± 2)**	Mean (SD)	−1.8 (0.79)	−0.3 (1.04)
	*p*-value, Two-Sample *t*-test	**0.0023**	-
	*p*-value, ANCOVA model	**0.0193**	-
Visit 4 **(Day 22 ± 2)**	Mean (SD)	−1.7 (1.06)	−0.4 (0.92)
	*p*-value, Two-Sample *t*-test	**0.0130**	-
	*p*-value, ANCOVA model	**0.0541**	-
Visit 5 **(Day 29 ± 2)**	Mean (SD)	−2.0 (1.05)	−0.3 (1.04)
	*p*-value, Two-Sample *t*-test	**0.0028**	-
	*p*-value, ANCOVA model	**0.0365**	-
Visit 6 **(Day 36 ± 3)**	Mean (SD)	−1.4 (1.43)	−0.1 (0.99)
	*p*-value, Two-Sample *t*-test	0.0482	-
	*p*-value, ANCOVA model	0.1909	
Visit 7 **(Day 43 ± 3)**	Mean (SD)	−1.4 (1.26)	−0.3 (1.28)
	*p*-value, Two-Sample *t*-test	0.0748	-
	*p*-value, ANCOVA model	0.2154	-
**Foreign Body Sensation**			
Visit 2 **(Day 8 ± 2)**	Mean (SD)	−1.9 (2.02)	−0.1 (0.99)
	*p*-value, Two-Sample *t*-test	0.0201	-
	*p*-value, ANCOVA model	0.5107	-
Visit 3 **(Day 15 ± 2)**	Mean (SD)	−1.6 (1.35)	−0.1 (0.99)
	*p*-value, Two-Sample *t*-test	0.0202	-
	*p*-value, ANCOVA model	0.3993	-
Visit 4 **(Day 22 ± 2)**	Mean (SD)	−2.3 (0.95)	−0.1 (0.83)
	*p*-value, Two-Sample *t*-test	**0.0001**	-
	*p*-value, ANCOVA model	**0.0107**	-
Visit 5 **(Day 29 ± 2)**	Mean (SD)	−2.3 (1.34)	0.1 (1.13)
	*p*-value, Two-Sample *t*-test	**0.0009**	-
	*p*-value, ANCOVA model	**0.0176**	-
Visit 6 **(Day 36 ± 3)**	Mean (SD)	−2.1 (1.37)	0.1 (1.13)
	*p*-value, Two-Sample *t*-test	**0.0020**	-
	*p*-value, ANCOVA model	**0.0409**	-
Visit 7 **(Day 43 ± 3)**	Mean (SD)	−2.2 (1.62)	0.5 (0.93)
	*p*-value, Two-Sample *t*-test	**0.0007**	-
	*p*-value, ANCOVA model	**0.0213**	-
**Dryness**			
Visit 2 **(Day 8 ± 2)**	Mean (SD)	−1.1 (1.10)	0.4 (1.30)
	*p*-value, Two-Sample *t*-test	**0.0191**	-
	*p*-value, ANCOVA model	**0.0443**	-
Visit 3 **(Day 15 ± 2)**	Mean (SD)	−0.9 (1.37)	0.3 (1.58)
	*p*-value, Two-Sample *t*-test	0.1177	-
	*p*-value, ANCOVA model	0.2546	-
Visit 4 **(Day 22 ± 2)**	Mean (SD)	−1.3 (1.06)	−0.3 (1.83)
	*p*-value, Two-Sample *t*-test	0.1462	-
	*p*-value, ANCOVA model	0.3445	-
Visit 5 **(Day 29 ± 2)**	Mean (SD)	−0.8 (1.69)	−0.3 (2.55)
	*p*-value, Two-Sample *t*-test	0.5899	-
	*p*-value, ANCOVA model	0.8980	-
Visit 6 **(Day 36 ± 3)**	Mean (SD)	−0.7 (1.16)	0.3 (1.67)
	*p*-value, Two-Sample *t*-test	0.1733	-
	*p*-value, ANCOVA model	0.3943	-
Visit 7 **(Day 43 ± 3)**	Mean (SD)	−0.6 (1.07)	0.0 (2.45)
	*p*-value, Two-Sample *t*-test	0.4946	-
	*p*-value, ANCOVA model	0.8554	-

Abbreviations: ANCOVA = Analysis of Covariance; ITT = intent-to-treat; SD = standard deviation. Note: N in headers represents the total number of subjects enrolled in each respective treatment group within the ITT population. Ora Calibra™ Ocular Discomfort and 4-Symptom Questionnaire assesses symptoms at the subject level. Grading ranges from 0 to 5. A score of 0 means no symptom. Baseline is defined as the last available pre-treatment measure. The bold is to emphasize the statistically significant data.

**Table 5 ijms-24-00554-t005:** Overall Summary of Adverse Events—Safety Population.

	0.1% RGN-259 (*n* = 10) *n* (%)	Placebo (*n* = 8) *n* (%)
Adverse Events (Ocular and Non-Ocular)		
Number of AEs	11	5
Number of Subjects with at Least One AE	4 (40.0%)	3 (37.5%)
Number of Treatment-Related AEs	1	0
Number of Subjects with at Least One Treatment-Related AE	1 (10.0%)	0
Number of SAEs	1	0
Number of Subjects with at Least One SAE	1 (10.0%)	0
Number of Treatment-Related SAEs	0	0
Ocular AEs		
Number of AEs	7	4
Number of Subjects with at Least One AE	3 (30.0%)	2 (25.0%)
Number of AEs in Treated Eyes	5	0
Number of Subjects with at Least One AE in a Treated Eye	2 (20.0%)	0
Number of AEs in Study Eyes	5	0
Number of Subjects with at Least One AE in a Study Eye	2 (20.0%)	0
Ocular AEs		
Number of AEs in Treated Fellow Eyes	0	0
Number of AEs in Untreated Fellow Eyes	2	4
Number of Subjects with at Least One AE in an Untreated Fellow Eye	2 (20.0%)	2 (25.0%)
Number of Treatment-Related AEs	1	0
Number of SAEs	0	0
Non-Ocular AEs		
Number of AEs	4	1
Number of Subjects with at Least One AE	2 (20.0%)	1 (12.5%)
Number of Treatment-Related AEs	0	0
Number of SAEs	1	0
Number of Subjects with at Least One SAE	1 (10.0%)	0

Abbreviations: AE = Adverse Event; SAE = Serious Adverse Event; TEAE = Treatment-Emergent Adverse Event. Note: *n* in the headers represents the total number of subjects enrolled in each respective treatment group within the Safety population. Percentages are based on the number of subjects in each treatment group. Treatment-related AEs are AEs with a relationship of possible, probable, definite, unclassified, or missing. All AEs collected in this study are treatment emergent. Note that if no AEs were observed in a category, then the information is not included.

**Table 6 ijms-24-00554-t006:** All Adverse Events—Safety Population.

	0.1% RGN-259 (*n* = 10)		Placebo (*n* = 8)	
System Organ Class (SOC) Preferred Term (PT)	Events	Subjects *n* (%)	Events	Subjects *n* (%)
Total—Ocular AEs	7	3 (30.0%)	4	2 (25.0%)
Eye disorders	7	3 (30.0%)	4	2 (25.0%)
Corneal epithelium defect	2	2 (20.0%)	0	0
Corneal opacity	2	1 (10.0%)	0	0
Keratic precipitates	1	1 (10.0%)	0	0
Visual impairment	1	1 (10.0%)	1	1 (12.5%)
Vitreous detachment	1	1 (10.0%)	0	0
Visual acuity reduced	0	0	3	1 (12.5%)
Total—Non-Ocular AEs	4	2 (20.0%)	1	1 (12.5%)
Infections and infestations			1	1 (12.5%)
Upper respiratory tract infection1			1	1 (12.5%)
General disorders and administration site conditions	1	1 (10.0%)		
Inflammation	1	1 (10.0%)		
Investigations	1	1 (10.0%)		
Blood glucose decreased	1	1 (10.0%)		
Nervous system disorders	1	1 (10.0%)		
Unresponsive to stimuli	1	1 (10.0%)		
Psychiatric disorders	1	1 (10.0%)		
Depression	1	1 (10.0%)		

## Data Availability

The data supporting these findings are available from the corresponding author upon reasonable request.

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
