# Peer review of "0.1% RGN-259 (Thymosin ß4) Ophthalmic Solution Promotes Healing and Improves Comfort in Neurotrophic Keratopathy Patients in a Randomized, Placebo-Controlled, Double-Masked Phase III Clinical Trial"

_ijms, 2022, doi:10.3390/ijms24010554_

Round 1

Reviewer 1 Report

This report suggest that low cost and use at room temperature 0.1% RGN-259(0.1% thymosin beta 4 topical ophthalmic solution) would be useful for NK treatment. It is interested.

I think Table 4 and line 145 to 151 should be deleted, because those were not objectively results.

Minor comments

Please insertion one page for Table 1 and 2, respectively.

Please do not separate title and table (Table 6).

Please insert the lines in Table 1 for easy to see.  

For example: between [White] and [Mackie Classification

            Between [Retinal detachment] and [Surgical Procedures]

            Between [Retinopexy] and [Area of Epithelial Defect]

Author Response

Response to reviewer #1

Comment: This report suggests that low cost and use at room temperature 0.1% RGN-259(0.1% thymosin beta 4 topical ophthalmic solution) would be useful for NK treatment. It is interested.

Response: We thank the reviewer for the careful comments on our manuscript.

Comment: I think Table 4 and line 145 to 151 should be deleted, because those were not objectively results.

Response. We agree that the data are subjective.  The data were collected by professional staff at each visit to improve the quality and be consistent.  Since the findings were significant, standardized, widely used for measuring ocular discomfort, and relative to the improvement in the NK patient QOL, we have decided to keep Table 4 and the information in the text.  We have added a clarification to the text in Results on the subjective nature of the data and also discussed this limitation in the Discussion.

Minor comments

Comment: Please insertion one page for Table 1 and 2, respectively.

Response: Due to the required minimum font size and journal template, this could not be done by us but hopefully the journal will be able to fix that once the manuscript is accepted.

Comment: Please do not separate title and table (Table 6).

Response: This problem has been fixed.

Comment: Please insert the lines in Table 1 for easy to see. For example: between [White] and [Mackie Classification Between [Retinal detachment] and [Surgical Procedures] Between [Retinopexy] and [Area of Epithelial Defect]

Response: Line spacing was added between each group.

Reviewer 2 Report

In the current study, authors have studied the efficacy and safety of 0.1% RGN-259 ophthalmic solution in the treatment of epithelial defects in patients. Authors have shown that subjects treated with 0.1% RGN-259 healed more complete and faster compared to placebo treatment. Overall, the study is thoroughly designed, the manuscript is well written, and data justifies the conclusion. The study is of significance to the field of ophthalmology.

Major limitation of the study is small sample size. As pointed out by the authors, treatment with 0.1% RGN-259 ameliorates epithelial defects as shown by primary efficacy endpoint but it was nominally significant. Similar is the case for other analysis. Conclusions may change with increase in sample size due to insufficient power of the study.

Minor corrections:

Occasional grammatical errors are present throughout the manuscript.

The results section should be segmented.

Multiple corrections should be employed for analyzing more than one ocular measurements. Simple Bonferroni correction will do.

Author Response

Comment: In the current study, authors have studied the efficacy and safety of 0.1% RGN-259 ophthalmic solution in the treatment of epithelial defects in patients. Authors have shown that subjects treated with 0.1% RGN-259 healed more complete and faster compared to placebo treatment. Overall, the study is thoroughly designed, the manuscript is well written, and data justifies the conclusion. The study is of significance to the field of ophthalmology.

Response: We thank the reviewer for careful analysis of our paper and the positive comments.

Comment: Major limitation of the study is small sample size. As pointed out by the authors, treatment with 0.1% RGN-259 ameliorates epithelial defects as shown by primary efficacy endpoint but it was nominally significant. Similar is the case for other analysis. Conclusions may change with increase in sample size due to insufficient power of the study.

Response: We agree that that major limitation with the study is the small sample size as mentioned in the second to last paragraph of the Discussion.  A larger study is planned.

Minor corrections:

Comment: Occasional grammatical errors are present throughout the manuscript.

Response: We apologize for the grammatical mistakes. The manuscript has been carefully edited for grammatical errors. 

Comment: The results section should be segmented.

Response:  The Results section is now segmented with titles and numbering,

Comment: Multiple corrections should be employed for analyzing more than one ocular measurements. Simple Bonferroni correction will do.

Response: We do not understand this request since only 2 groups were analyzed for one hypothesis concerning the treatment effect on a single primary endpoint.

Round 2

Reviewer 2 Report

Authors have satisfactorily addressed the reviewer comments.